# Synthesis, Characterization and Photocatalytic Activity of N-doped Cu₂O/ZnO Nanocomposite on Degradation of Methyl Red

**Yohannes Teklemariam Gaim \***, **Gebrekidan Mebrahtu Tesfamariam,**
**Gebretinsae Yeabyo Nigussie and Mengstu Etay Ashebir**

Department of Chemistry, College of Natural and Computational Sciences, Mekelle University, Mekelle P.O.Box: 231, Ethiopia; gebrekidanmebrahtu11102007@gmail.com (G.M.T.); g.tinsae21@gmail.com (G.Y.N.); mengstu.etay@mu.edu.et (M.E.A.)

**\*** Correspondence: yohannesteklemariam1@gmail.com; Tel.: +25-19-3203-4081

**Abstract:** In this study, a N-doped Cu₂O/ZnO nanocomposite was prepared by a co-precipitation and thermal decomposition technique from CuCl₂, 2H₂O, ZnSO₄, 7H₂O and CO(NH₂)₂ as precursors. The as-synthesized nanocomposites were characterized using X-ray diffraction (XRD), scanning electron microscopy (SEM), Fourier transform infrared analysis (FT–IR) and an ultraviolet–visible (UV–Vis) reflectance spectrometer. From the XRD diffractogram of N-doped Cu₂O/ZnO nanocomposite, cubic and hexagonal wurtzite crystal structures of Cu₂O, and ZnO, respectively were identified. The UV-vis reflectance spectra illustrated that the absorption edge of N-doped Cu₂O/ZnO nanocomposite is more extended to the longer wavelength than ZnO, Cu₂O and Cu₂O/ZnO nanomaterials. FT–IR bands confirmed the presence of ZnO, Cu₂O, and nitrogen in the N-doped Cu₂O/ZnO nanocomposite. Photocatalytic activity of the as-synthesized nanocomposite was tested for methyl red degradation using sunlight as an energy source by optimizing the concentration of the dye and amount of the catalyst loaded. The degradation efficiency was greater in N-doped Cu₂O/ZnO nanocomposite as compared to ZnO, Cu₂O and Cu₂O/ZnO nanomaterials. This is due to the coupling of the semiconductors which increases the absorption and exploitation capability of solar light and increases the charge separation as well. Besides that, nitrogen doping can extend absorption of light to the visible region by decreasing the energy gap. Therefore, N-doped Cu₂O/ZnO nanocomposite is a solar light-active photocatalyst which can be used in the degradation of organic pollutants.

**Keywords:** photocatalyst; nanocomposite; dye degradation; co-precipitation

---

## 1. Introduction

Population growth and rapid urbanization/industrialization are among the greatest causes of environmental pollution and consumption of a large amount of energy [1]. Synthetic organic dyes that are used in various industries such as pharmaceuticals, textiles, cosmetics, paper, and plastic factories have led to severe environmental pollution as a result of their discharging contaminated and colored wastewater into the water stream [2]. They adversely affect the quality of water, prevent light penetration and diminish photosynthetic reactions. Moreover, some dyes are both poisonous and cancer-causing [3]. To address the above issues, several treatment methods have been commonly used. The treatment techniques including adsorption, chemical precipitation and coagulation show reduced efficiency and generate other pollutants like toxic gases and slurry that require additional purification [4,5]. Therefore, advanced technology-based treatments have been suggested for the removal of these pollutants.

Advanced oxidation processes (AOPs) have attracted much attention as a substitute for traditional treatment routes for the removal of toxic organic pollutants into harmless products. AOPs have benefits such as degradation of organic pollutants to green products and capability of operating at normal temperature and pressure [6]. Among AOPs, heterogeneous photocatalysis is an emerging technique, which is valuable for environmental and energy applications. It is a photochemical reaction which takes place on the surface of the solid catalyst and encompasses oxidation from photogenerated holes and reduction from photogenerated electrons at the same time [7]. $TiO_2$, ZnO, CdS, ZnS and $Fe_2O_3$ are some of the outstanding semiconductors used as photocatalysts [8]. Among these, $TiO_2$ and ZnO are the top applicable as photocatalysts [9,10]. The energy levels of these semiconductors are nearly comparable. However, ZnO is easily obtainable; it absorbs a large portion of solar light and has great photocatalytic performance than $TiO_2$ [11,12].

Zinc oxide-based materials are used in the area of multifunctional electrode for both energy conversion and storage applications, like lithium-ion batteries and Dye-sensitized Solar Cells [13], gas sensors, monitoring air quality and optical devices due to its exceptional properties, for example, being inexpensive, photoconductive response, pyroelectricity and surface functionalization [14], high binding energy and electron mobility [15]. This metal oxide based semiconductor also has great application in the field of optoelectronic devices such as light-emitting diodes, flat panel displays, transparent semiconductors and conductive oxides, due to its good optical properties [16]. Different research investigated sensing and photovoltaic applications of ZnO-based materials such as ammonia gas sensing using Ag/ZnO flower and Cu-doped ZnO nanostructures [15,17], tin-doped ZnO thin films as a $NO_2$ gas sensor [18], ZnO@$In_2O_3$ coreshell nanofibers for ethanol vapor sensing [19], ZnO-based quaternary transparent conductive oxide materials for solar cells [16], and CdO-ZnO nanocones for efficient electrode materials [18]. The application of lithium ion batteries has been a major success in small electronic devices. However, the shortage of lithium resources challenges its application in large-scale electrical energy storage systems [20]. The Layered Sodium Transition-Metal Oxides are promising materials that can minimize the challenges of lithium batteries due to excellent cyclic stability and rate performance which significantly contribute to the development of large-scale electrical energy storage systems [21].

However, ZnO absorbs only in the ultraviolet section of electromagnetic radiation for the reason that its bandgap energy is large. As a result, its photocatalytic activity is low under solar radiation as the ultraviolet (UV) constituent of solar energy that touches the Earth is only 3–5% [22,23]. Nowadays, the existing photocatalysts such as ZnO are modified by doping or co-doping with metals and non-metals to enrich their photocatalytic activity [24,25]. Moreover, compositing of dissimilar nanostructured semiconductors develops their photocatalytic performance by sharing of their charge carriers to each other [26].

Cuprous oxide ($Cu_2O$) is a narrow bandgap semiconductor which has been thought of as a possible visible light photocatalyst. Electrons of $Cu_2O$ can undergo a transition from the valence band to the conduction band using visible light as a source of energy. However, the photo-induced electrons and holes recombine within microseconds after their generation, which can influence its photocatalytic action negatively. Up to the present time, graphene and selected metals were coupled with $Cu_2O$ to delay the recombination of the photoinduced electrons and holes. $Cu_2O$ joined with large bandgap metal oxides such as ZnO is expected as an operational means to control the problem of recombining the charge carriers [27]. In this work, we synthesized a N-doped $Cu_2O$/ZnO nanocomposite via co-precipitation and thermal decomposition methods and tested its photocatalytic activity in the degradation of methyl red (Scheme 1), which is considered as model dye pollutant.

**Scheme 1.** Structure of methyl red.

## 2. Experimental

### 2.1. Chemicals

The chemicals used were: cupric chloride dihydrate, zinc sulfate heptahydrate, sodium hydroxide, ascorbic acid, urea, $C_2H_6O$, methyl red, and distilled water. The chemicals were analytical grade and used throughout without further purification. Distilled water was used in all the experiments.

### 2.2. Synthesis of Photocatalysts

The undoped $Cu_2O$ nanoparticles were prepared from $CuCl_2 \cdot 2H_2O$ as a precursor via precipitation and reduction techniques. 0.01 moles of $CuCl_2 \cdot 2H_2O$ was dissolved in 100 mL deionized water. The solution was adjusted in its pH level by adding 30 mL of 2 M NaOH drop by drop with continuous stirring until 12.2. During the addition of the NaOH aqueous solution, a deep blue precipitate was formed quickly. After being stirred for 30 min, 2.2 g of $C_6H_8O_6$ was added into the above solution to reduce $Cu^{2+}$ into $Cu^+$ and stirred for an extra 30 min. The color of the precipitate then changed to brick red indicating the formation of $Cu_2O$. The brick-red precipitate was filtered and washed four times using deionized water and dried at 100 °C in an oven [27]. To show high crystallinity and remove organic impurities, the powders were calcined at 500 °C in a furnace for 2 h.

ZnO nanoparticles were prepared using $ZnSO_4 \cdot 7H_2O$ as a precursor through a precipitation method [27]. To prepare ZnO nanoparticles, 0.01 moles of $ZnSO_4 \cdot 7H_2O$ dissolved in 100 mL deionized water. While stirring, 30 mL of 2 M NaOH was dropped carefully into the aqueous solution of zinc sulfate until the pH of the solution reached 12.4. Upon addition of NaOH aqueous solution, a white precipitate was formed and stirred for 30 min. The white product was separated by filtration and washed using deionized water repeatedly to remove impurities and dried at 100 °C in an oven. Finally, the powders were calcined at 500 °C in a furnace for 2 h.

The $Cu_2O$/ZnO nanocomposite was synthesized from $CuCl_2 \cdot 2H_2O$ and $ZnSO_4 \cdot 7H_2O$ via the co-precipitation method followed by reduction using ascorbic acid; 0.01 moles of $CuCl_2 \cdot 2H_2O$ and 0.01 moles of $ZnSO_4 \cdot 7H_2O$ were dissolved in 100 mL deionized water and stirred using a magnetic stirrer; 30 mL of 2 M NaOH dropped slowly into the mixture of cupric chloride and zinc sulfate aqueous solutions with continuous stirring and the pH of the solution became 11.8. A light blue precipitate was formed and stirred for 30 min. Once the solution has been stirred for half an hour, the copper reduced from +2 to +1 oxidation states after 2.2 g of $C_6H_8O_6$ added while stirring for an additional 30 min. The color of the precipitate changed from light blue into yellow as soon as the ascorbic acid was added as a reducing agent. Lastly, the yellow precipitate was filtered, washed, repeatedly, using deionized water and ethanol, and dried at 100 °C in the oven [27]. The nanocomposite was then calcined at 500 °C in a furnace for 2 h.

To prepare the N-$Cu_2O$/ZnO nanocomposite, 2.681 g of CO $(NH_2)_2$ dissolved in ethanol and 25 g of uncalcined $Cu_2O$/ZnO nanocomposites was added to the solution. The mixture was stirred up to mix well and then dried. Finally, the powder was calcined at 500 °C for 2 h and the nitrogen-doped $Cu_2O$/ZnO nanocomposite was obtained [25].

### 2.3. Characterization Techniques

The following were used as characterization techniques: X-ray diffraction (XRD) analysis was conducted using X-ray diffractometer (Shimadzu XRD-7000, Shimadzu Corp., Kyoto, Japan) with a Cu-K$\alpha$ radiation ($\lambda = 0.15406$ nm), step scan mode with step time and degree ($2\theta$) of 0.4 s and 0.02°, respectively, for the range of 10° to 80°), to know crystal structures and average crystallite size of the samples. The surface morphology of the materials was examined using a scanning electron microscope (SEM), a JEOL JSM-5610 (JEOL, Ltd., Akishima, Tokyo) equipped with an Everhart-Thornley detector. The chemical compositions of the prepared samples were characterized using Spectrum 65 Fourier transform infrared (FT–IR) (PerkinElmer, Waltham, MA, USA) in the range 4000–400 cm$^{-1}$ using KBr pellets. The optical property of the nanocomposites was recorded using a PerkinElmer Lamda 35 spectrometer which is operated at a wavelength range of 200–800 nm and the absorption spectra have been obtained from reflectance data using the Kubelka–Munk algorithm [28].

### 2.4. Photocatalytic Activities

The photocatalytic activities of the as-synthesized nanomaterials were tested on the degradation of methyl red dye (Scheme 1) as a model pollutant under sunlight as an energy source at ambient temperature. The degradation experiments were performed as follows: 500 L glass vessels containing 60 mg/L solution of methyl red and appropriate amount of nanomaterials (18 mg/L) were stirred under a dark for 1 h to attain absorption–desorption equilibrium before irradiating. All experiments were carried out under direct solar irradiation explicitly sunny days in between 10 AM and 2 PM when the solar intensity variations were insignificant [29]. Thereafter, the mixed solution was exposed to sunlight and 8 mL of the heterogeneous mixture was withdrawn within 20 min intervals of irradiating to sunlight, centrifuged at 3000 rpm for 6 min and filtered using filter paper to separate the catalyst before conducting the absorption measurement. The pure supernatant liquid was evaluated for methyl red dye concentration after measuring the absorbance at 410 nm using an ultraviolet–visible (UV–Vis) spectrophotometer and the percentage degradation of methyl red solution was calculated according to the following equation:

$$\% \text{ degradation } = \frac{A_0 - A_t}{A_0} \times 100\% \tag{1}$$

where $A_0$ is initial absorbance of methyl red dye and $A_t$ is absorbance of methyl red at each time interval "t".

## 3. Results and Discussion

### 3.1. X-ray Diffraction (XRD) Patterns of the Nanomaterials

The diffraction patterns of $Cu_2O$, ZnO, $Cu_2O$/ZnO, and N-doped $Cu_2O$/ZnO nanoparticles are shown in Figure 1. In the XRD patterns of ZnO, $Cu_2O$/ZnO and N-$Cu_2O$/ZnO nanomaterials, diffraction peaks appeared at $2\theta$ = 31.76°, 34.40°, 36.24°, 47.53°, 56.59°, 62.85°, 66.37°, 67.90°, 69.07° which correspond to (100), (002), (101), (102), (110), (103), (200), (112), and (201) planes of the hexagonal wurtzite structure of zinc oxide. Similar results were reported in [30]. The diffraction peaks in the XRD patterns of $Cu_2O$, $Cu_2O$/ZnO and N-$Cu_2O$/ZnO at $2\theta$ value of 29.57°, 36.40°, 42.32°, 61.43°, 73.55° and 77.40° correspond to the reflection from (110), (111), (200), (220), (311) and (222) crystal planes of the cubic structure of cuprous oxide which is in agreement with [31]. There is no other diffraction peak displayed from impurities such as CuO, $Cu(OH)_2$ and $Zn(OH)_2$, indicating the purity of the nanostructured materials. As shown in Figure 1d, no additional peak was displayed due to nitrogen doping which might be nitrogen introduced into the ZnO [32] and $Cu_2O$ lattices without changing their crystal structures. The insignificant shift in the diffraction peaks of N-$Cu_2O$/ZnO nanocomposite corresponds to the possibility of substituting oxygen by nitrogen. This can be mainly attributed to the highest resemblance among nitrogen and oxygen atoms in terms of electronegativity and atomic radius.

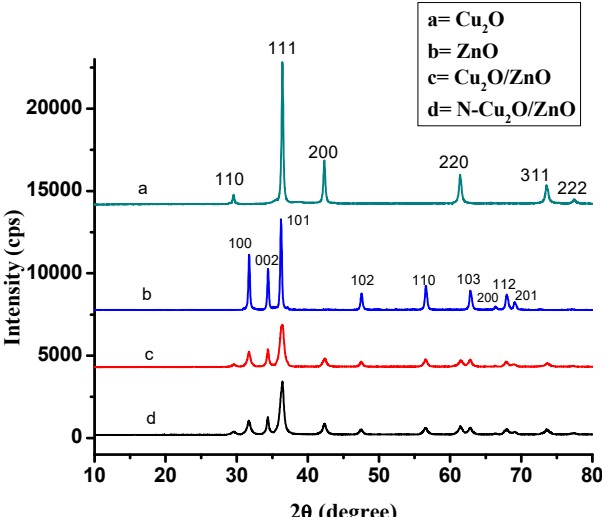

**Figure 1.** X-ray diffractogram of (a) Cu$_2$O, (b) ZnO, (c) Cu$_2$O/ZnO and (d) N-doped Cu$_2$O/ZnO nanomaterials.

The average crystallite sizes of the nanomaterials were calculated from the intensive peak using the Scherer equation:

$$D = \frac{0.9\lambda}{\beta \, \cos\theta} \qquad (2)$$

where D is the crystallite size, λ is the wavelength, θ is the Bragg angle and β is the full width at half maximum in radian. The average crystallite sizes of ZnO, Cu$_2$O, Cu$_2$O/ZnO and N-Cu$_2$O/ZnO were found at 33.72 nm, 32.33 nm, 14.15 nm, and 13.57 nm, respectively. Based on these results, the size of the crystallites is decreased in N-doped Cu$_2$O/ZnO nanocomposite compared to the Cu$_2$O and ZnO. The effect of decreasing the crystallite size may be ascribed to the insertion of nitrogen (incorporation of dopant) in Cu$_2$O and ZnO lattices [32,33] and this was also confirmed in the photocatalytic degradation experiment. This insertion of nitrogen into the Cu$_2$O and ZnO lattices can disturb the growth process of the particles, which might be the reason for the reduction of crystallite size in N-Cu$_2$O/ZnO nanocomposite compared to Cu$_2$O and ZnO nano-level particles.

### 3.2. Scanning Electron Microscopy (SEM) Analysis

Surface morphology of ZnO, Cu$_2$O, Cu$_2$O/ZnO and N-Cu$_2$O/ZnO nanomaterials was determined by SEM as shown in Figure 2a–d, respectively. From SEM micrographs, it was evident that the morphology of ZnO showed some agglomerated nanoparticles with irregular morphology, which is in line with [30]. However, SEM micrographs of Cu$_2$O, Cu$_2$O/ZnO and N-Cu$_2$O/ZnO samples were relatively ordered and showed that the agglomerations of particles were much less (Figure 2b,d) than in ZnO NPs with the nanocrystals, which had a truncated octahedron shape; this might be due to the presence of Cu$_2$O [34,35].

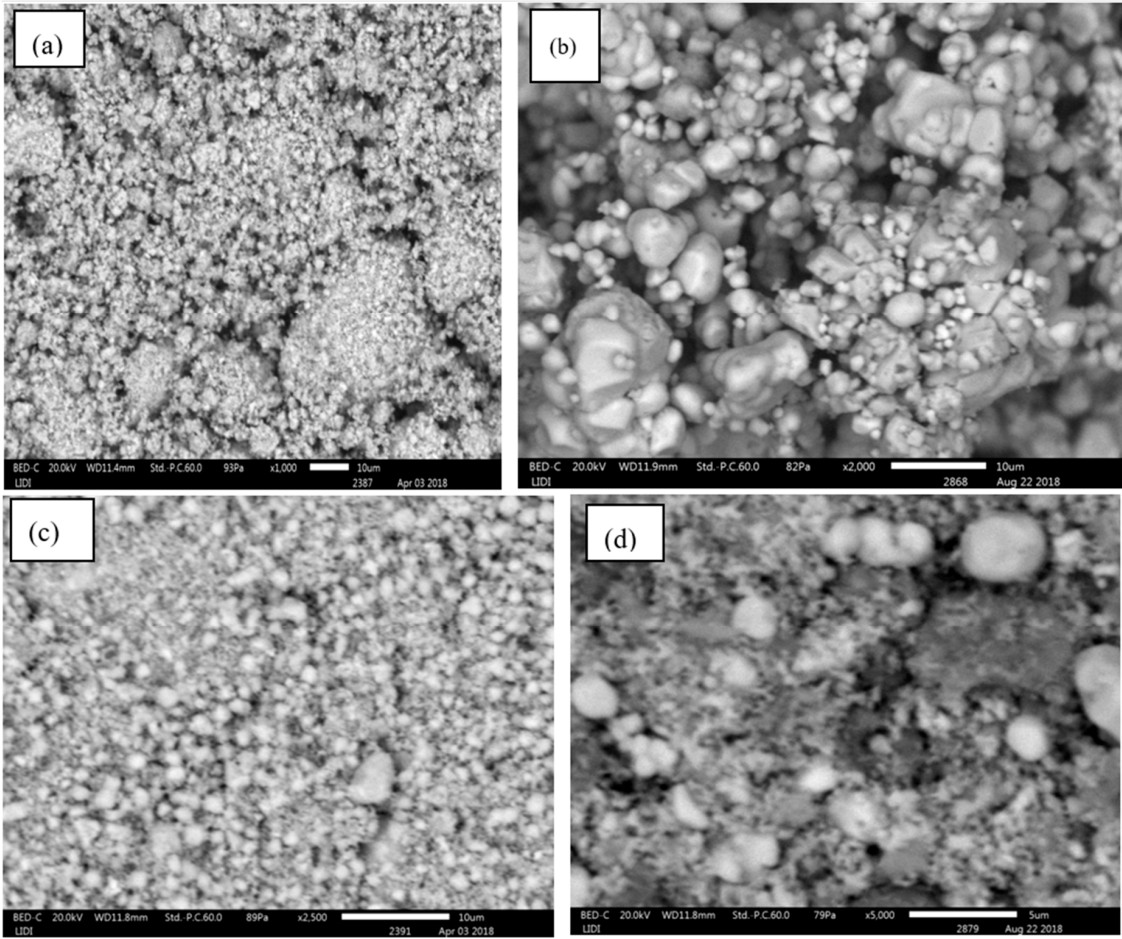

**Figure 2.** Scanning electron microscopy (SEM) morphology of (**a**) ZnO, (**b**) Cu$_2$O, (**c**) Cu$_2$O/ZnO and (**d**) N-doped Cu$_2$O/ZnO nanomaterials.

*3.3. Fourier Transform Infrared (FT–IR) Analysis*

The FT–IR bands of ZnO, Cu$_2$O, Cu$_2$O/ZnO and N-doped Cu$_2$O/ZnO nanomaterials are given in Figure 3. The peaks appeared in the range of 490–505 cm$^{-1}$ in all the samples except in Cu$_2$O and correspond to the stretching vibration of ZnO, which agreed with the findings of previous studies [36,37]. In the FT–IR spectra of Cu$_2$O, Cu$_2$O/ZnO and N-doped Cu$_2$O/ZnO nanoparticles, there were peaks in the range of 610–630 cm$^{-1}$, which corresponds to the stretching vibration of Cu$_2$O; a similar result was reported by [34]. The peak that appeared at 3169 cm$^{-1}$ in the FT–IR spectrum of the N-doped Cu$_2$O/ZnO nanocomposite might be due to N-H stretching vibration mode. The band located at 1441 cm$^{-1}$ might be due to N-H bending vibration mode [38]. Besides the above absorption bands, there was an additional peak located at 431 cm$^{-1}$ in the FT–IR spectrum of the N-doped Cu$_2$O/ZnO nanocomposite, which might be attributed to the metal–nitrogen (M–N) stretching vibration similar results have been reported [32]. Therefore, the bands indicated the presence of ZnO, Cu$_2$O and nitrogen in the N-doped Cu$_2$O/ZnO nanocomposite.

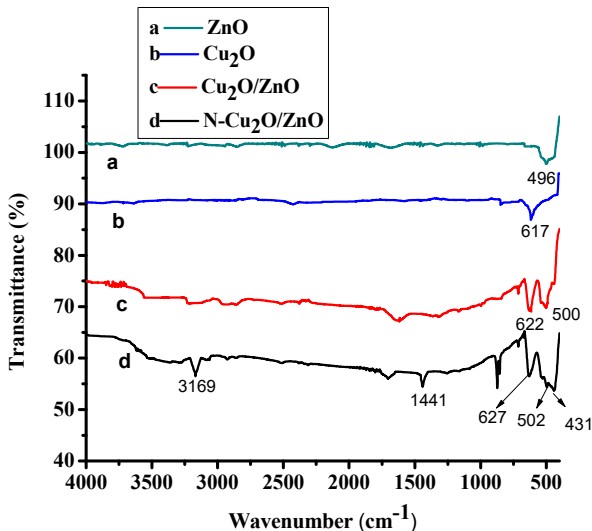

**Figure 3.** Fourier transform infrared (FT–IR) spectrums of (a) ZnO, (b) $Cu_2O$, (c) $Cu_2O/ZnO$ and (d) N-$Cu_2O/ZnO$ nanomaterials.

### 3.4. Optical Property Analysis

The region of absorption of electromagnetic radiation of the prepared nanomaterials was studied using UV–Vis diffuse reflectance spectroscopy. The spectra for the absorption of light by ZnO, $Cu_2O$, $Cu_2O/ZnO$ and N-$Cu_2O/ZnO$ are given in Figure 4. ZnO, $Cu_2O$, $Cu_2O/ZnO$ and N-$Cu_2O/ZnO$ absorbed at 360, 440, 470 and 500 nm, respectively. For ZnO, the absorption edge was in the ultraviolet region. The $Cu_2O/ZnO$ composite nanoparticles absorbed light in the visible region because of the presence of $Cu_2O$ [27]. The absorption edges were extended into 470 nm by coupling ZnO with $Cu_2O$ and then further to 500 nm by doping with nitrogen.

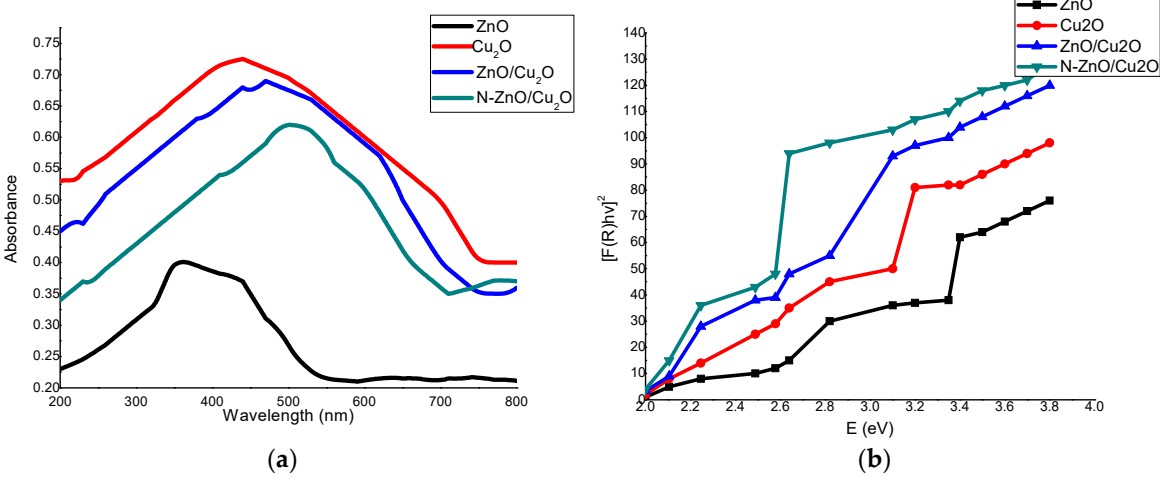

（**a**）　　　　　　　　　　　　　　　　　　　（**b**）

**Figure 4.** (**a**) Spectra for the optical property of ZnO, $Cu_2O$, $Cu_2O/ZnO$ and N-$Cu_2O/ZnO$ nanomaterials. (**b**) Kubelka–Munk function versus energy plots of ZnO, $Cu_2O$, $ZnO/Cu_2O$, and N-$ZnO/Cu_2O$ nanoparticles.

The bandgap energy of the nanocomposites can be inferred by extrapolation of the linear portion of the graph between the modified Kubelka–Munk function $[F(R)hν]^2$ versus photon energy(hν) [39]; as shown in Figure 4b. The bandgap energy of ZnO and $Cu_2O$ is 3.4 eV and 2.81 eV, respectively; however, there is decrement for $ZnO/Cu_2O$ (2.64 eV) and N-$ZnO/Cu_2O$ (2.48 eV). The greatly extended absorption of light by N-$Cu_2O/ZnO$ to the visible region may be ascribed to the creation of a new energy level above the valence band of $Cu_2O$ and ZnO as a result of nitrogen doping, leading to

narrowing of the bandgap to the visible region for harvesting more photons in the sunlight, which is in agreement with previous findings [40].

### 3.5. Photocatalytic Activity

#### 3.5.1. Optimization of N-Cu$_2$O/ZnO Loading

To know the optimal value of catalyst dosage, a series of photocatalytic degradation experiments were conducted by varying the amount of the N-Cu$_2$O/ZnO nanocomposite from 120 to 240 mg/L as optimized by [27]. The photocatalytic degradation of methyl red under sunlight irradiation using a different amount of N-doped Cu$_2$O/ZnO nanocomposite is shown in Figure 5. The figure shows that, as the amount of catalyst loading increases from 120 mg/L to 180 mg/L, the degradation efficiency also increased. The reason is that increasing the catalyst loading increases the surface area and quantity of reaction sites on the surface of the photocatalyst. Consequently, the amount of hydroxyl radical formation increases too, which enables the photocatalytic degradation of the dye. However, the degradation efficiency was decreased when the amount of the catalyst dosage was beyond 180 mg/L. This might be due to light-scattering and screening effects [41,42]. Besides, agglomeration also occurs when the concentration of catalyst is high; which results in the decreasing of catalyst surface area and causes diminishing of degradation efficiency [42,43]. Therefore, the degradation efficiency was greatest when the amount of the catalyst used was 180 mg/L and was the optimal value in the experiment.

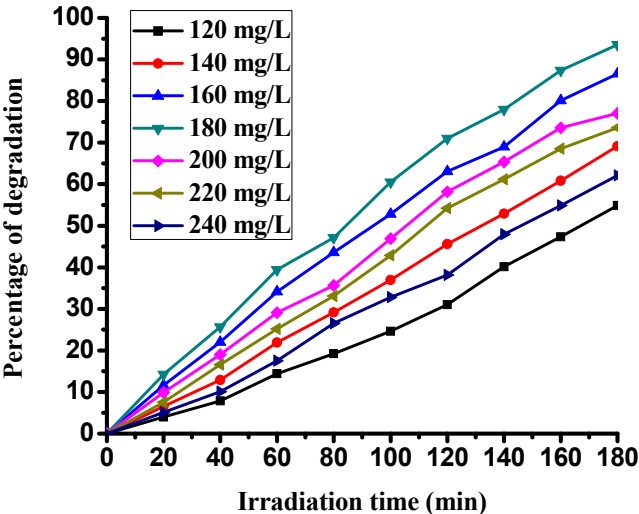

**Figure 5.** Optimization of N-Cu$_2$O/ZnO nanocomposite loading for the degradation of methyl red under sunlight.

#### 3.5.2. Optimization of Methyl Red Concentration

Once the amount of the catalyst was optimized at 180 mg/L, a series of photocatalytic degradation experiments were conducted by varying the concentration of methyl red from 40 to 100 mg/L as adjusted by [27] to know the proper amount of the dye. As shown in Figure 6, as the concentration of the dye was enlarged from 40 mg/L to 60 mg/L, the degradation efficiency was likewise enhanced. However, as the concentration of the dye goes above 60 mg/L, the degradation efficiency sharply decreased. The reason for the decreasing of photocatalytic degradation efficiency with increasing concentration of the dye is that the higher dye concentration could affect the transmission of light which leads to a decrease in hydroxyl radical formation. The total amount of active sites on the surface of the catalyst was limited by the amount of catalyst loaded. Therefore, in the solution having fixed catalyst dosage, an inadequate amount of hydroxyl radicals that can attack the methyl red can form, hence leading to the diminishing of degradation capability [43].

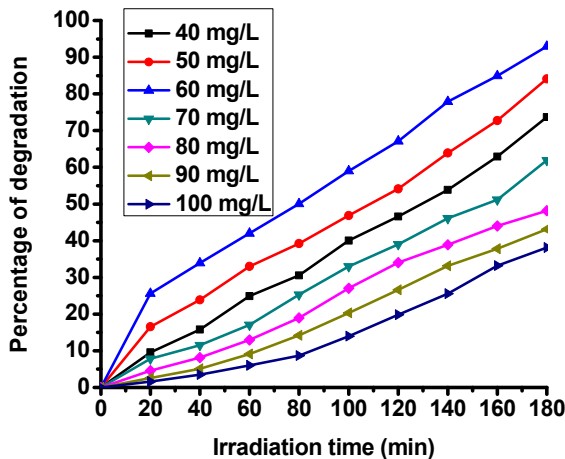

**Figure 6.** Optimization of the initial concentration of methyl red on the photocatalytic degradation using N-Cu$_2$O/ZnO nanocomposite under sunlight.

### 3.5.3. Evaluation of the Photocatalytic Activities of Cu$_2$O, ZnO, Cu$_2$O/ZnO and N-Cu$_2$O/ZnO under the Optimized Catalyst Amount and Dye Concentration

The photocatalytic performance of Cu$_2$O, ZnO, Cu$_2$O/ZnO, and N-Cu$_2$O/ZnO nanomaterials were evaluated in the degradation of methyl red dye, as shown in Figure 7. The photocatalytic degradation efficiency of methyl red reached 45.5%, 54%, 84.5% and 93.5% using Cu$_2$O, ZnO, Cu$_2$O/ZnO, and N-doped Cu$_2$O/ZnO, respectively, within 180 min irradiation time. Among these nanomaterials, the N-doped Cu$_2$O/ZnO nanocomposite displayed better photocatalytic activity than the others under the optimized conditions. The activity of N-doped Cu$_2$O/ZnO nanocomposite is enhanced because of the formation of a heterojunction [35]. The p-Cu$_2$O/n-ZnO heterojunction might significantly increase the absorption and exploitation capability of solar light; the electrons transfer from the one semiconductor to the other encourages the charge separation and construct significant synergistic effect in the degradation of the dye [35]. In addition to the effect of the coupling of the two semiconductors, the improved photocatalytic efficiency of N-Cu$_2$O/ZnO nanocomposite is due to doping with nitrogen. Incorporating non-metals, for example nitrogen, can diminish the energy gap and extend absorption of light to the visible region of electromagnetic radiation [44]. In other words, nitrogen can modify the energy levels of both Cu$_2$O and ZnO nanoparticles. Besides the above reasons, the enhanced photocatalytic activity of N-doped Cu$_2$O/ZnO composite nanoparticles may be due to the reduction in particle size and creation of defect sites.

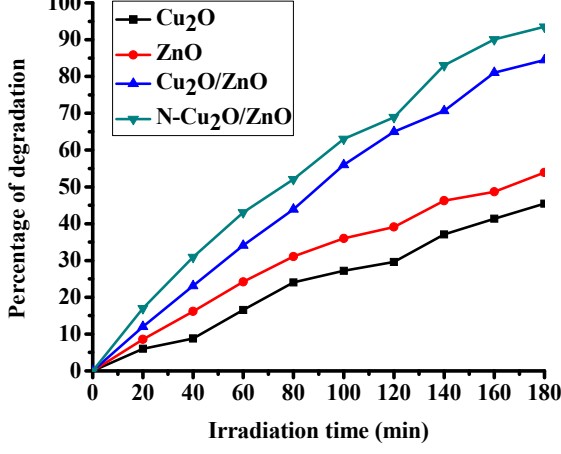

**Figure 7.** Evaluation of photocatalytic properties of ZnO, Cu$_2$O, Cu$_2$O/ZnO and N-doped Cu$_2$O/ZnO on the degradation of methyl red under solar light.

### 3.5.4. Proposed Mechanism

Scheme 2 demonstrates the photocatalytic degradation mechanism of methyl red by N-doped $Cu_2O$/ZnO composite nanoparticles under sunlight irradiation. When the semiconductors are exposed to sunlight, a transition of electrons from the valence band to the conduction band and formation of holes in the valence band (Equations (3) and (4)) can take place. The conduction band of cuprous oxide is at higher position compared to zinc oxide (Jiang et al., 2013), the photo-induced electrons in the $Cu_2O$ conduction band can simply transfer to the ZnO conduction band Equation (5), which can successfully prevent the recombination of charge carriers. The electrons at the conduction band of $Cu_2O$ and ZnO undergo a reaction with adsorbed oxygen to give a peroxide radical anion Equation (6). It is impossible to oxidize the $OH^-$ by holes of $Cu_2O$ because the valence band edge of $Cu_2O$ is higher in a position [45]. However, the holes of ZnO oxidize the hydroxyl ion to yield a hydroxyl radical Equation (8), a strong oxidizing agent that can break down the organic dye. Besides, the peroxide radical anion undergoes a reaction with the hydrogen ion to produce $HO_2$ and $H_2O_2$ Equation (9). The hydrogen peroxide is then reacted with the peroxide radical anion to create the powerful hydroxyl radicals Equation (11). Eventually, the hydroxyl radical oxidizes the dye into photocatalytic degradation products Equation (12) [35,43]. The reaction steps are described below.

$$Cu_2O \ + \ h\upsilon \ \rightarrow \ Cu_2O\left(h^+\right) \ + \ Cu_2O\ (e^-) \tag{3}$$

$$ZnO \ + \ h\upsilon \ \rightarrow \ ZnO\left(h^+\right) \ + \ ZnO\ (e^-) \tag{4}$$

$$Cu_2O\ (e^-) \ + \ ZnO \ \rightarrow \ Cu_2O \ + \ ZnO\ (e^-) \tag{5}$$

$$Cu_2O\ (e^-)/ZnO\ (e^-) \ + \ O_{2\text{-}} \tag{6}$$

$$H_2O \ \rightarrow \ H^+ \ + \ OH^- \tag{7}$$

$$ZnO\left(h^+\right) \ + \ OH^- \ \rightarrow \ OH \tag{8}$$

$$O_{2\text{-}} \ + \ H^+ \rightarrow \ HO_2 \tag{9}$$

$$HO_2 \ + \ HO_2 \ \rightarrow O_2 \ + \ H_2O_2 \tag{10}$$

$$H_2O_2 \ + \ O_{2\text{-}} \ \rightarrow \ OH^- \ + \ OH \ + \ O_2 \tag{11}$$

$$OH \ + \ Dye \ \rightarrow \ H_2o + CO_2 \tag{12}$$

The schematic illustration for the photocatalytic degradation mechanism of methyl red under sunlight is summarized below.

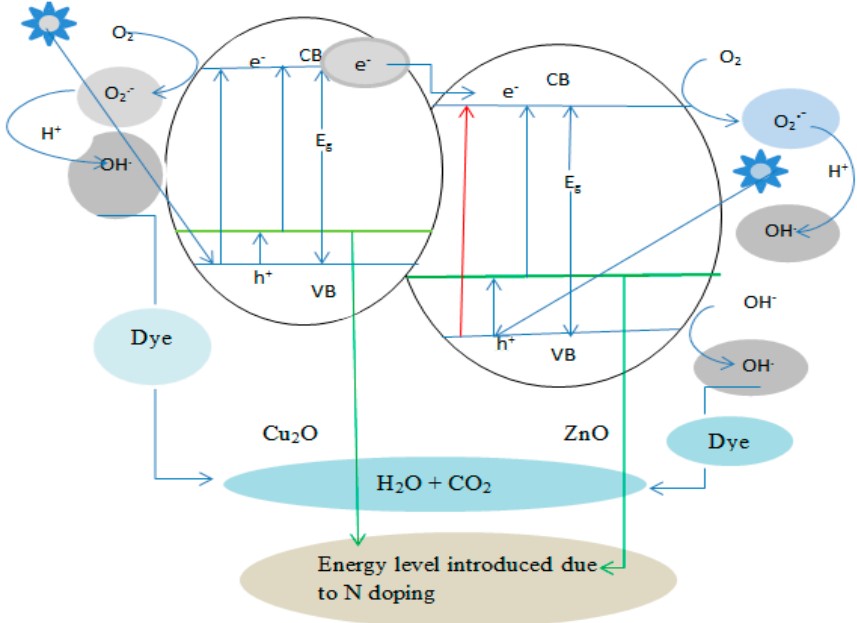

**Scheme 2.** Schematic illustration of the photocatalytic degradation of methyl red using N-Cu₂O/ZnO nanocomposite under sunlight illumination.

### 3.6. Kinetic Study of the Photocatalytic Decolorization

The kinetics of the photocatalytic degradation of methyl red over ZnO, Cu$_2$O, Cu$_2$O/ZnO and N-doped Cu$_2$O/ZnO is shown in Figure 8. The plots of $\ln(C_0/C_t)$ versus irradiation time indicated linear curves. The linearity of the kinetic curve shows that the photocatalytic degradation of methyl red follows pseudo-first-order kinetics. A fairly good correlation coefficient value to the pseudo-first-order kinetics of $R^2 > 0.96$ was obtained, which is in agreement with other reports [46]. The rate constants (reaction rate) were predicted from the slope of the graph. The rate constants are found to be 0.00338, 0.00423, 0.01005 and 0.01088 min$^{-1}$ for Cu$_2$O, ZnO, Cu$_2$O/ZnO, and N-Cu$_2$O/ZnO nanomaterials, respectively. The rate constants of the photocatalytic degradation of methyl red using N-Cu$_2$O/ZnO and Cu$_2$O/ZnO nanocomposites are larger than pure Cu$_2$O and ZnO nanoparticles. Therefore, the photocatalytic activities of N-Cu$_2$O/ZnO and Cu$_2$O/ZnO nanocomposites based on the rate constants are found to be greater than Cu$_2$O and ZnO nanoparticles [5,47].

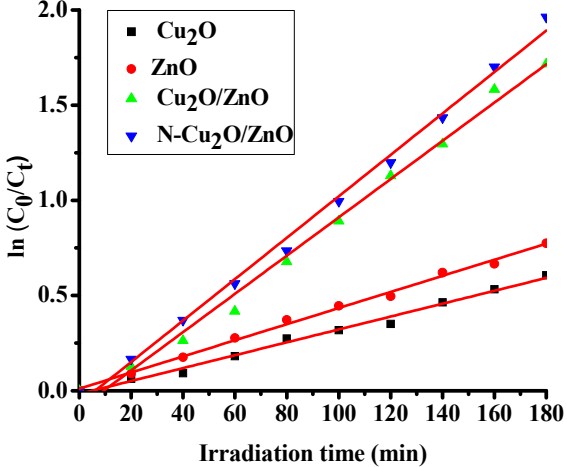

**Figure 8.** Reaction kinetics curve.

## 4. Conclusions

N-doped $Cu_2O/ZnO$ nanocomposite was synthesized by co-precipitation and thermal decomposition methods. The absorption edge of the nanocomposite was more extended to the visible region of electromagnetic radiation compared to ZnO, $Cu_2O$ and $Cu_2O/ZnO$ nanomaterials. The photocatalytic activity of N-doped $Cu_2O/ZnO$ nanocomposite has enhanced efficiency than ZnO, $Cu_2O$ and $Cu_2O/ZnO$ nanomaterials. The enhancement is due to the coupling of $Cu_2O$ and ZnO semiconductors, which can drop the recombination rate of the charge carriers, and improve the absorption and utilization ability to sunlight. In addition to that, doping the nanocomposite with nitrogen can outspread the photoabsorption to longer wavelengths. Based on the findings, N-doped $Cu_2O/ZnO$ nanocomposite is an effective solar light active photocatalyst that can be used for the removal of organic dyes from wastewater.

**Author Contributions:** All the authors participated in conceptualizing, executing, analyzing, editing, and reviewing this article.

**Funding:** This research received no external funding.

**Conflicts of Interest:** The authors declared no conflict interest.

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
