# Peer review of "Synthesis, Characterization and Photocatalytic Activity of N-doped Cu2O/ZnO Nanocomposite on Degradation of Methyl Red"

_jcs, doi:10.3390/jcs3040093_

Round 1
Reviewer 1 Report
The paper can be published after the following minor revision:
1) The photoreactor configuration for tests under solar light should be better described.
2) The linear portion of Kubelka-Munk diagrams used for the calculus of band-gap should be better evidenced.
Author Response
We are very thankful for careful and thorough reading of this manuscript and for the constructive comments and suggestions, which will help to improve the quality of this manuscript. The comments are regarding to photoreactor configuration for tests under solar light and linear portion of Kubelka-Munk diagrams used for the calculus of band-gap are addressed are in the attached document.
The initial layout has also modified based on your constructing ideas. Please see the authors’ response in red Colour below to comments.

Reviewer 2 Report
This manuscript proposes the synthesis, characterization and photocatalytic activity of N-doped Cu2O/ZnO nanocomposite on degradation of methyl red. The topic is interesting, and certainly consistent with the contents to be proposed to the readers of “Journal of Composites Science”. However, the manuscript is not so well written and should be improved to be read with pleasure: this represents an important aspect in the current scenario of publications in international journals. Overall, I think that this manuscript could be accepted if the Authors will be able to take into account the following major revisions (in terms of bibliographic updates, grammar corrections and content deepening):
Detailed revisions: I spent several hours reading this manuscript, and Authors are asked to follow carefully the attached PDF file where I highlighted some points to be addressed. The attached file also contains language mistakes and typos (they are many in this work and should not be present when submitting a manuscript to an international journal: Authors are asked to check the manuscript better next time); some questions related to manuscript contents could also be present and Authors must consider them properly before submitting the revised manuscript. A point-by-point reply is required when the revised files are submitted. Considering the amount of mistakes and typos present in this manuscript, a further check carried out by a native English speaker or by a professional English language center is suggested. The Introduction should give a wider overview on the present scenario related to current trends in ZnO-based materials, both in terms of recently published reviews and research articles. In particular, application in sensing and photovoltaics are missing and a paragraph on this topic is highly suggested to be added in the Introduction. Authors are invited to go through the literature published in the last six months on these issues, and also on concepts developed some years ago in this field. Some of them are also mentioned in the above mentioned PDF file. Authors should provide a clear explanation on the experimental error of the proposed research work. In particular, reproducibility of the phenomena described in the manuscript should be clearly stated in the “Results and Discussion” section; besides, some notes in the “Materials and Methods” section should be added highlighting which kind of experimental approach has been followed to check the reproducibility of the proposed system, the latter being of noteworthy importance in the present research field. References: an article submitted to a journal should be consistent with the contents that it typically proposes in its table of contents. However, by checking the references of this manuscript, I did not find any articles published in this journal: this sounds rather strange. Maybe, Authors could check better the topics recently addressed by this journal, studying its table of contents and enriching the Introduction (as mentioned above) with some articles connected to this field.
Author Response
Dear Reviewer
Thank you very much for reviewing our manuscript. We also greatly appreciate the reviewers for their complimentary comments and suggestions. We sincerely apologies for the great time it has taken us to respond to these comment particularly, the grammar and punctuation and hope that a revised version of the manuscript will still be considered. We have modified the paper in response to the extensive and insightful reviewer comments. We have added additional explanations on sunlight irradiation, Application of ZnO based materials in sensing and photovoltaics and modified the whole part of the reference that exclude irrelevant references. Please find attached initial layout a point-by-point response to reviewer’s concerns indicated by red color. We hope that you find our responses satisfactory and that the manuscript is now acceptable for publication.
Sincerely

Round 2
Reviewer 2 Report
The manuscript has been amended in many parts, but the bibliography is still questionable since adequate referencing to previous applications of ZnO-based materials in light-driven technologies is not sufficient. In my previous review, I attached a PDF file that contained, at page 2, useful references to fix this issue, but the Authors did not consider them. The file is attached again. Minor revisions, therefore, are still requested.
